# Cohort profile: Genetic data in the German Socio-Economic Panel Innovation Sample (SOEP-G)

**Philipp D. Koellinger**[1]*, **Aysu Okbay**[1], **Hyeokmoon Kweon**[1], **Annemarie Schweinert**[2], **Richard Karlsson Linnér**[1,3], **Jan Goebel**[4], **David Richter**[5,6], **Lisa Reiber**[7], **Bettina Maria Zweck**[8], **Daniel W. Belsky**[9,10], **Pietro Biroli**[11], **Rui Mata**[7,12], **Elliot M. Tucker-Drob**[13], **K. Paige Harden**[13], **Gert Wagner**[5,7,14], **Ralph Hertwig**[7]

1 Department of Economics, School of Business and Economics, Vrije Universiteit Amsterdam, Amsterdam, The Netherlands, 2 Department of Economics, University of Wisconsin-Madison, Madison, Wisconsin, United States of America, 3 Department of Economics, Leiden Law School, Leiden University, Leiden, The Netherlands, 4 German Socio-Economic Panel Study, Deutsches Institut für Wirtschaftsforschung (DIW Berlin), Berlin, Germany, 5 Educational Science and Psychology, Freie Universität Berlin, Berlin, Germany, 6 SHARE Berlin, Berlin, Germany, 7 Center for Adaptive Rationality, Max-Planck Institute for Human Development, Berlin, Germany, 8 Kantar Public, München, Germany, 9 Department of Epidemiology and Butler Columbia Aging Center, Mailman School of Public Health, Columbia University, New York, New York, United States of America, 10 PROMENTA Center, University of Oslo, Oslo, Norway, 11 Department of Economics, University of Bologna, Bologna, Italy, 12 Faculty of Psychology, University of Basel, Basel, Switzerland, 13 Department of Psychology and Population Research Center, University of Texas at Austin, Austin, Texas, United States of America, 14 Federal Institute for Population Research, Wiesbaden, Germany

* p.d.koellinger@vu.nl

**Data Availability Statement:** The collected phenotypes from all SOEP samples can be accessed via user agreements with DIW Berlin (https://www.diw.de/en/diw_01.c.601584.en/data_

## Abstract

The German Socio-Economic Panel (SOEP) serves a global research community by providing representative annual longitudinal data of respondents living in private households in Germany. The dataset offers a valuable life course panorama, encompassing living conditions, socioeconomic status, familial connections, personality traits, values, preferences, health, and well-being. To amplify research opportunities further, we have extended the SOEP Innovation Sample (SOEP-IS) by collecting genetic data from 2,598 participants, yielding the first genotyped dataset for Germany based on a representative population sample (SOEP-G). The sample includes 107 full-sibling pairs, 501 parent-offspring pairs, and 152 triads, which overlap with the parent-offspring pairs. Leveraging the results from well-powered genome-wide association studies, we created a repository comprising 66 polygenic indices (PGIs) in the SOEP-G sample. We show that the PGIs for height, BMI, and educational attainment capture 22 ~ 24%, 12 ~ 13%, and 9% of the variance in the respective phenotypes. Using the PGIs for height and BMI, we demonstrate that the considerable increase in average height and the decrease in average BMI in more recent birth cohorts cannot be attributed to genetic shifts within the German population or to age effects alone. These findings suggest an important role of improved environmental conditions in driving these changes. Furthermore, we show that higher values in the PGIs for educational attainment and the highest math class are associated with better self-rated health, illustrating complex relationships between genetics, cognition, behavior, socio-economic status, and health. In summary, the SOEP-G data and the PGI repository we created provide a valuable

access.html, contact email soepmail@diw.de). DIW Berlin shares the genetic PCs and all PGIs constructed for the SOEP-G in a standard phenotype file. This data version also includes an indicator for individuals who did not pass the strict QC pipeline. This allows users to decide whether to conduct their analyses using the full sample for which PGIs were constructed or the slightly smaller set that passed strict QC. DIW Berlin will also share family relationship data for each related pair, inferred from both the survey and genetic data, which will also contain genetic kinship estimates. The raw genetic data from SOEP-G will be stored on the European Genome-Phenome Archive (https://ega-archive.org/) from 2024 onwards, and DIW Berlin will handle data access applications (contact email soepmail@diw.de). Raw genetic data must be stored on high-security servers that meet the technical and organizational security measures required by the General Data Protection Regulation of the European Union.

**Funding:** The data collection was supported by the German Research Foundation (Leibniz Prize to RH), a European Research Council Consolidator Grant (647648 EdGe to PK), the Jacobs Foundation (EMTD, KPH, DWB), National Institute of Health/National Institute of Child Health and Human Development grant R01HD092548 (KPH), the NORFACE DIAL Grant 462-16-100 (PB), the University of Basel (RM), and the Canadian Institute for Advanced Research (DWB) and the Max Planck Institute for Human Development (GGW). The Population Research Center at the University of Texas at Austin (KPH and ETD) is funded by NIH Center grant P2CHD04284. The funders had no role in study design, data collection and analysis, decision to publish, or manuscript preparation.

**Competing interests:** The authors have declared that no competing interests exist.

resource for studying individual differences, inequalities, life-course development, health, and interactions between genetic predispositions and the environment.

## Introduction—Why was this cohort set up?

Almost all human traits are partly heritable, including health outcomes, personality, and behavioral tendencies [1, 2]. All properties that make us unique as individuals are to some degree affected by random genetic variation within and between families. Moreover, genetic and environmental causes of individual differences are interrelated. For example, environmental conditions can affect how genetic differences between individuals translate into differences in socio-economic and health outcomes [3–5]. And, genetic differences among people manifest in trait differences partly via environmental channels, for example, via genetically influenced personal interests that lead to self-selection into specific environments and reinforcement mechanisms consisting, for instance, of behaviors of parents, teachers, peers, or colleagues [6, 7]. Importantly, the fact that genetic differences are linked to differences in behavior and health does not imply simplistic biological determinism and puts no upper bound on the relevance of the environment or the possibilities for intervention [8, 9].

The heritability [10, 11] of behavioral, psychological, and economic phenotypes (e.g., educational attainment, personality, risk preference) and health outcomes (e.g., cardiovascular disease, dementia) typically range between 30% and 70%, with, based on twin studies, an average estimated heritability of 49% ($SE$ = 0.004) across all traits [2]. Thus, a substantial amount of variation in outcomes that epidemiologists and behavioral scientists study can be statistically linked to genetic differences among people. Ignoring genetics would imply that a substantial source of individual differences would remain unexamined, potentially leading to biased estimations that could prompt wrong and possibly counterproductive conclusions [12].

Twin studies also suggest that environmental factors are important for social scientific outcomes and a broad variety of diseases [2]. Thus, detailed information about living conditions, attitudes, and behavior could inform health-related research questions. However, most medical research datasets only contain basic information about these factors, limiting the possibility of fully understanding their importance for health outcomes [13].

While genetically informed study designs are already common in medical research and have yielded numerous important insights into disease mechanisms [14, 15], genetic data in the behavioral and social sciences is still relatively rare [16]. Nevertheless, integrating genetic data into the research of the behavioral and social sciences (*e.g.*, economics, psychology, sociology, political science) opens up new possibilities to (i) control for genetic confounders that are otherwise unknown and that may lead to biased empirical results, (ii) increase the statistical power of empirical analyses by absorbing residual variance in multiple regression analyses, yielding smaller standard errors of the estimated parameters, (iii) study the interactions of genetic factors and environmental exposures, (iv) use random genetic differences among individuals to identify causal pathways, and (v) better understand how social (dis)advantages are transmitted across generations and how parents, peers, teachers, and policymakers can potentially alleviate or amplify such (dis)advantages [16, 17]. Thus, integrating genetic data into the behavioral and social sciences offers researchers new tools to study key questions to reach more robust inferences based on their empirical analyses, as illustrated by several recent examples [18–20].

The genetic underpinnings of behavior, socio-economic outcomes, and health often overlap. For example, educational attainment has substantial genetic correlations with smoking

(-0.3), lung cancer (-0.4), obesity (-0.2), Alzheimer's disease (-0.3), and longevity (+0.6) [16, 21], illustrating the complex relationships between components of genetic variation, human behavior, environmental conditions, and health outcomes.

These considerations motivated us to collect genetic data in the Innovation Sample of the German Socio-Economic Panel Study (SOEP-IS), to contribute additional value to the already existing and widely known interdisciplinary and longitudinal SOEP data set that is accessible and frequently used by the global scientific community [22]. Adding genetic data to this sample opens many new research opportunities for the medical and social-science research community.

SOEP-IS was started in 2011 as an addition to the SOEP-Core sample, which provides representative annual data of private households in Germany since 1984 [23]. Similar to the SOEP-Core sample, SOEP-IS is a valuable data resource for researchers who want to explore long-time societal changes, relationships between early life events and later life outcomes; interdependencies between the individual and the family or household; mechanisms of intergenerational mobility and transmission; accumulation processes of resources; short- and long-term effects of institutional change and policy reforms; and migration dynamics [23]. Besides containing a set of basic questions identical to the SOEP-Core, the SOEP-IS longitudinal panel survey incorporates innovative content that is purely user-designed, including measurements that go beyond the scope of standardized questionnaire formats.

As a household study, the SOEP-IS typically contains data about all household members, including many mother-father-child trios, parent-offspring duos, childhood development, parenting practices, and family dynamics. Furthermore, due to the sampling method and longitudinal nature of the data, the available phenotypes in the SOEP-IS span all stages of life—from the (pre-)natal stage, early childhood, adolescence, adulthood, all the way to retirement and the end of life (see Fig 1). We refer to the genotyped subsample of the SOEP-IS as the SOEP-G sample.

Already existing genotyped cohorts in Germany (e.g., BASE-II [24], DHS [25], HNRS [26], KORA [27], SHIP [28]) focus on specific health outcomes or are limited in scope to specific regions or age groups. Thus, as of now, SOEP-G is the only genotyped dataset that is based on a representative sample of households in Germany, and that contains family data as well as a rich array of longitudinal information about health, personality, family dynamics, living conditions, attitudes, and socio-economic behaviors and outcomes. This makes the sample particularly valuable to study long-term developments and the intergenerational transmission of inequalities in health and well-being. Furthermore, the sample is ideally suited to study the impact of environmental conditions unique to Germany, such as specific public policies and changes therein or the potential consequences of German reunification. Fig 2 shows the geographic distribution of genotyped households in the SOEP-G sample, illustrating the sample's coverage of all German states and metropolitan areas (e.g., Berlin, Hamburg, Munich, Ruhr region).

To enable the collection of genetic data in the SOEP-IS, we established a research consortium of scientists from Germany (Max-Planck Institute for Human Development, German Institute of Economic Research), the Netherlands (Vrije Universiteit Amsterdam), Switzerland (University of Zurich, University of Basel), and the USA (University of Texas at Austin, Columbia University). The consortium was spearheaded by Philipp Koellinger (Vrije Universiteit Amsterdam) and Ralph Hertwig (Max-Planck Institute for Human Development). Koellinger's team in Amsterdam developed and guided the data collection procedures, processed the collected genetic data, and generated polygenic indices for public use.

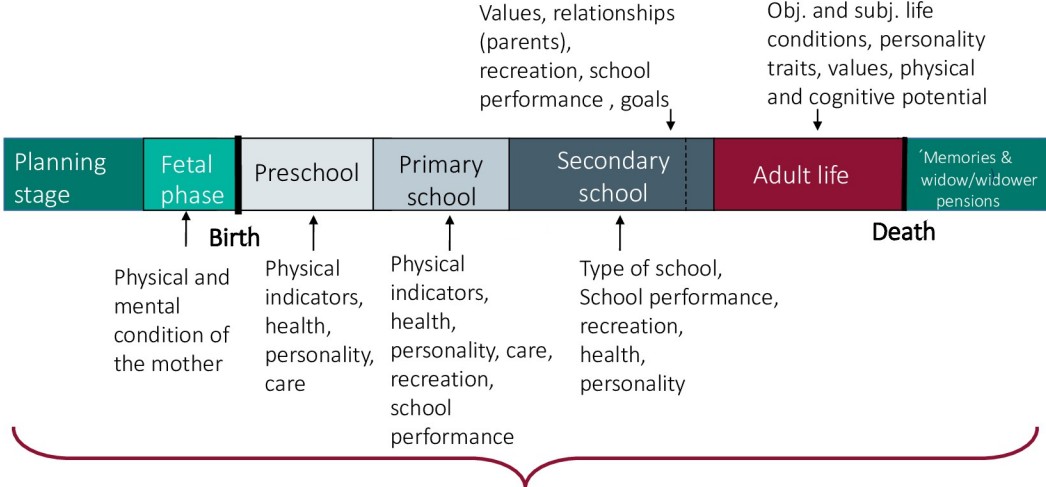

**Fig 1. Life course perspective of the SOEP-IS sample.**

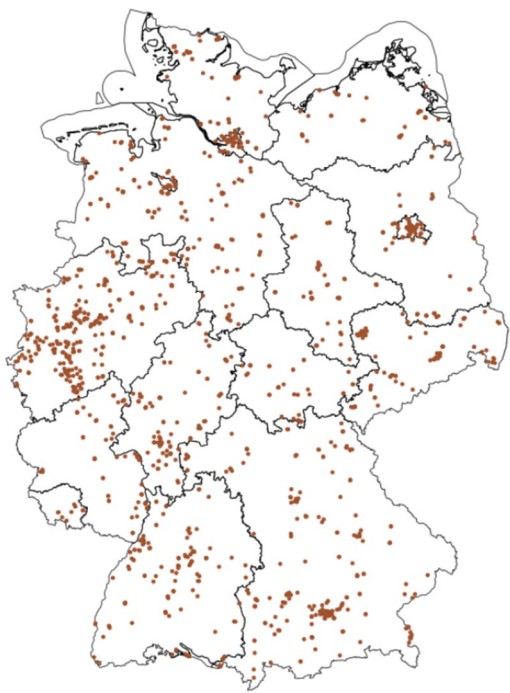

**Fig 2. Geographic distribution of genotyped households in the SOEP-G sample. Note**: Own illustration based on spatial datasets from Reference [29] under data license Germany–attribution–Version 2.0 dl-de/by20 (https://www.govdata.de/dl-de/by-2-0).

## Materials and methods—Who is in the sample?

The sampling and interviewing methods, as well as the baseline characteristics of the sample, were previously described in detail [22, 23]. In short, SOEP-IS is based on a random sample of households in Germany. Annual computer-assisted personal interviews are conducted face-to-face, and information is collected on the household- and individual levels (e.g., individual and household incomes). The main survey instrument is a household questionnaire being answered by the head of the household. In addition, there is an individual questionnaire that each household member age 17 and older is supposed to answer. The obtained information usually covers the current situation (e.g., family composition or satisfaction with life), but in some contexts, it includes the past (e.g., job changes and employment biographies) and the future (e.g., expected life satisfaction in 5 years, and the chance of re-employment).

The main guardian (the caretaker, usually the mother) is asked about their children younger than 17. If members of an originally sampled household leave the household, (e.g., because of a divorce or children forming their own household), the original and the split household are interviewed. The comprehensive tracing rules, covering all individuals who (even temporarily) lived in SOEP households, represent a comparative advantage of SOEP compared to other household panel surveys. They allow users to track household dynamics and their implications at the household and individual levels. To maintain a reasonable sample size and to address panel attrition, refreshment samples of the residential population of Germany were integrated in 2012, 2013, 2014, and 2016.

The precondition for participation in the SOEP-G—as part of SOEP-IS 2019—was that the person or child lives in a participating household. 6,576 people were originally invited to participate in SOEP-IS 2019, 1,074 of whom were children. Not everyone takes part every year, and there are always people who move away, die, or do not want to participate in the survey anymore. Therefore, of the original sample, 4,283 persons who were at least 17 years old (i.e., persons of survey age), as well as 875 children and youths (<17 years of age) lived in a participating household in 2019. 2,598 individuals provided a valid genetic sample, including 215 children and teenagers. A requirement for an offspring of at most 17 years of age to participate in collecting genetic data was that both guardians agreed. The valid genetic samples were sent from the survey company Kantar GmbH to the Human Genomics Facility (HuGe-F) at the Erasmus Medical Center in Rotterdam for analysis.

Despite attrition, the SOEP-G sample, when compared with census data ([www.destatis.de](http://www.destatis.de)), is very similar to the German population in terms of age ($Mean_{census}$ = 52 years vs. $Mean_{SOEP-G}$ = 55 years), sex (51% $Female_{census}$ vs. 54% $Female_{SOEP-G}$), and region (20% East $Germany_{census}$ vs. 19% East $Germany_{SOEP-G}$). However, residents without German citizenship are under-represented in the SOEP-G sample (12% census vs. 4% SOEP-G).

Participants who agreed to donate DNA are similar to the overall SOEP-IS sample regarding socio-demographics, subjective health ratings, and life satisfaction (see Table 1). However, the comparison to the overall SOEP-core sample shows that the participants in the SOEP-IS sample are overall socioeconomically better off than those in the core sample and potentially the overall German population. Nonetheless, representativeness means that the survey covers all groups of persons. Unequal participation rates are compensated for in descriptive figures by weighting the cases; accordingly, weighting is unnecessary for analyses of subgroups with different sampling probabilities and for multivariate analyses.

Parents were more hesitant to enroll their offspring (<17 years of age) than themselves to collect genetic data. Compared to an overall consent rate of 58% (2,496 out of 4,282 valid interviews), only 26% of the eligible offspring participated in the collection of genetic data (228 out of 875). Importantly, however, offspring for whom genetic data was collected closely resemble

**Table 1. Descriptive statistics of the SOEP-G adult sample ($\geq$ 17 years old).**

| | Total (Core) | | Total (IS) | | Interview | | Consent | | Genotyped | | Polygenic Indices Created | |
|---|---|---|---|---|---|---|---|---|---|---|---|---|
| | **Mean** | **SD** | **Mean** | **SD** | **Mean** | **SD** | **Mean** | **SD** | **Mean** | **SD** | **Mean** | **SD** |
| Age | 44.1 | 17.9 | 53.7 | 19.1 | 54.7 | 18.4 | 55.1 | 19.1 | 55.4 | 19.0 | 55.2 | 19.0 |
| Sex (% female) | 50.1 | 50.0 | 52.8 | 49.9 | 53.1 | 49.9 | 53.8 | 49.9 | 54.0 | 49.9 | 54.6 | 49.8 |
| East Germany (% yes) | 18.4 | 38.7 | 20.0 | 40.0 | 20.1 | 40.1 | 19.5 | 39.6 | 19.4 | 39.5 | 19.7 | 39.8 |
| German (% yes) | 68.2 | 46.6 | 94.9 | 22.1 | 95.8 | 20.1 | 96.4 | 18.7 | 96.3 | 18.8 | 97.3 | 16.2 |
| Partnered (% yes) | 45.1 | 49.8 | 40.9 | 49.2 | 40.9 | 49.2 | 40.1 | 49.0 | 39.6 | 48.9 | 41.0 | 49.2 |
| School degree: low (% yes) | 48.0 | 50.0 | 37.9 | 48.5 | 37.9 | 48.5 | 39.7 | 48.9 | 39.6 | 48.9 | 37.8 | 48.5 |
| School degree: high (% yes) | 26.6 | 44.2 | 31.2 | 46.3 | 31.2 | 46.3 | 29.2 | 45.5 | 29.0 | 45.4 | 29.7 | 45.7 |
| Employment (% yes) | 58.1 | 49.3 | 53.3 | 49.9 | 53.3 | 49.9 | 50.9 | 50.0 | 50.9 | 50.0 | 51.3 | 50.0 |
| Net Income | 1807.1 | 1940.8 | 1959.1 | 1303.9 | 1959.1 | 1303.9 | 1922.4 | 1300.5 | 1914.7 | 1258.4 | 1925.3 | 1261.4 |
| Subjective Health (1–5) | 3.5 | 1.0 | 3.3 | 1.0 | 3.3 | 1.0 | 3.3 | 1.0 | 3.3 | 1.0 | 3.3 | 1.0 |
| Life Satisfaction (0–10) | 7.4 | 1.8 | 7.5 | 1.7 | 7.5 | 1.7 | 7.6 | 1.7 | 7.6 | 1.7 | 7.6 | 1.7 |
| Heavy drinking | 2.2 | 14.7 | 4.4 | 20.5 | 4.4 | 20.5 | 4.5 | 20.8 | 4.6 | 20.9 | 4.5 | 20.7 |
| Smoking (% yes) | 16.3 | 37.0 | 24.2 | 42.9 | 24.3 | 42.9 | 24.1 | 42.8 | 23.6 | 42.5 | 23.9 | 42.7 |
| Height (cm) | 171.6 | 9.6 | 171.4 | 9.4 | 171.4 | 9.4 | 171.2 | 9.5 | 171.2 | 9.5 | 171.4 | 9.6 |
| BMI | 26.4 | 5.2 | 26.6 | 5.1 | 26.6 | 5.1 | 26.9 | 5.3 | 26.9 | 5.3 | 26.8 | 5.1 |
| Observations | 45,306 | | 5,502 | | 4,283 | | 2,496 | | 2,375 | | 2,076 | |

Note: The column names indicate: **Total** every individual in the given SOEP sample, **Interview** individuals who responded to 2019–2020 wave survey, **Consent** individuals who gave consent for genotyping, **Genotyped** individuals successfully genotyped, **Polygenic indices created** individuals of European genetic ancestry who passed quality control and whose polygenic indices were created.

the overall sample of offspring in the sample in terms of age, sex, geographic location, and citizenship, as well as parental characteristics (see Table 2).

## Materials and methods—What has been measured?

### Phenotypes

The SOEP-IS [22, 30] contains a set of core questions that are identical to about 44% of the questions asked in the SOEP-Core survey [23], including variables such as age, gender, height, weight, education, employment status, income, life satisfaction, personality, living conditions, attitudes, preferences, and occupational classifications following the International Standard Classification of Occupations (ISCO). In addition, the SOEP-IS contains a broad range of short-term experiments and longer-term surveys that were not (yet) evaluated to be suitable to the SOEP-Core survey (yet) because they pose a higher risk of refusal and panel attrition or because they deal with very specific research issues. Every year, researchers can propose new survey modules or experiments for inclusion in the SOEP-IS. The SOEP management team and the SOEP survey committee then select which modules will be included in the next survey wave [22]. The SOEP-IS innovation modules also act as a test bed for how respondents react. Some particularly important and successful modules (e.g., risk preference) can later be integrated into the much larger SOEP-Core survey, which collects data from ∼ 15,000 households comprising ∼ 26,000 individuals per year, including ∼ 3,000 children and youths.

Health outcomes in the SOEP-IS are primarily measured by self-reporting doctor diagnoses for various diseases, subjective evaluations of health and well-being, doctor visits, and the need for care. Furthermore, dried blood samples were tested for SARS-CoV-2 antibodies and oral-nasal swabs for viral RNA in a part of the SOEP-IS sample between Oct 2020 and Feb 2021,

**Table 2. Descriptive statistics of children and adolescents (<17 years old) in the SOEP-G sample.**

| | Total | | Consent | | Genotyped | | Polygenic Indices Created | |
|---|---|---|---|---|---|---|---|---|
| | Mean | SD | Mean | SD | Mean | SD | Mean | SD |
| Age | 8.4 | 4.8 | 8.6 | 4.9 | 8.7 | 4.9 | 8.6 | 4.9 |
| Sex (% female) | 49.2 | 50.0 | 49.6 | 50.1 | 49.8 | 50.1 | 48.9 | 50.1 |
| East Germany (% yes) | 18.1 | 38.5 | 19.3 | 39.6 | 18.1 | 38.6 | 19.9 | 40.0 |
| German (% yes) | 96.0 | 19.6 | 96.5 | 18.4 | 96.3 | 19.0 | 99.4 | 7.5 |
| Father Age at birth | 34.2 | 5.4 | 34.2 | 5.4 | 34.2 | 5.4 | 34.2 | 5.5 |
| Mother Age at birth | 30.8 | 5.4 | 30.9 | 5.4 | 30.9 | 5.4 | 30.9 | 5.2 |
| Father School degree: low (% yes) | 30.3 | 46.1 | 30.5 | 46.2 | 30.5 | 46.2 | 27.2 | 44.7 |
| Father School degree: high (% yes) | 39.4 | 49.0 | 38.9 | 48.9 | 38.9 | 48.9 | 40.4 | 49.3 |
| Mother School degree: low (% yes) | 34.4 | 47.6 | 34.6 | 47.7 | 34.6 | 47.7 | 31.6 | 46.7 |
| Mother School degree: high (% yes) | 29.0 | 45.5 | 28.6 | 45.3 | 28.6 | 45.3 | 27.2 | 44.6 |
| Father Net Income | 2846.3 | 1522.1 | 2848.3 | 1528.2 | 2848.3 | 1528.2 | 2917.1 | 1599.8 |
| Mother Net Income | 1433.9 | 930.7 | 1426.6 | 930.9 | 1426.6 | 930.9 | 1425.4 | 925.8 |
| Observations | 1,074 | | 228 | | 215 | | 176 | |

Note: See Table 1's note for the column name definitions.

providing opportunities to study factors influencing infections with SARS-CoV-2 and long-term consequences [31].

Furthermore, the SOEP-IS allows users to add anonymized spatial information (e.g., federal states, spatial planning regions, counties, municipalities, and postal codes, as well as GPS coordinates) and can be linked to administrative records from the German Pension Insurance and the Employer-Employee Study [23, 32].

An overview of the SOEP-IS survey content and examples of modules is provided in Box 1. The complete questionnaire of the 2019 survey wave, the 2019 SOEP annual report, and a description of all SOEP-IS modules from 2011–2018 are available online [33–35]. An online companion for the entire data collection is available (http://companion-is.soep.de/).

## Genetics

DNA was extracted from saliva samples that were collected using Isohelix IS SK-1S buccal swabs with Dri-Capsules. Genotyping was performed using Illumina Infinium Global Screening Array-24 v3.0 BeadChips, yielding raw data for 2,598 individuals and 725,831 variants, of which 688,618 were autosomal.

The genotype missingness rate was greater than 5% in 484 individuals. Further analyses revealed that the high missingness rates for these individuals were largely driven by interviewer effects, possibly due to not following the sample collection protocol accurately, including incorrect use of (or entirely missing) DriCapsules that slow down the decay of DNA, low saliva and DNA yield, or polluted samples (see sections 2 and 3 in S1 File).

Since we expect that the vast majority of analyses in the genotyped SOEP-IS data will rely on polygenic indices (PGIs) [36] rather than single genetic variant analyses, we implemented two different quality control (QC) pipelines, mild-QC and strict-QC, that are described in detail in the S1 File. The mild-QC pipeline yields a higher sample size and both QC protocols yield approximately equally predictive PGIs (see below and section 7 in S1 File). Depending on the research questions investigators will address, either the mild-QC or the strict-QC data can be used to maximize the statistical power of the analyses.

Box 1. Summary of SOEP-IS survey content by topics and examples of modules.

1. *Demography and Population*

Country of origin, birth history

2. *Work and Employment*

Change of job, contractual working hours, employment status, evening and weekend work, financial compensation for overtime, industry sector and occupational classification, job search, leaving a job, maternity / parental leave, registered unemployed, self-employment reasons, side jobs, supervisory position, use of professional skills, vacation entitlement, work from home, work time regulations, workload

3. *Income, Taxes, and Social Security*

Asset balance, benefits and bonuses from employer, financial support received, individual gross / net income, inheritances, pension plans, social security, wage tax classification, alimony, household income and expenses, investments, repayments of loans

4. *Family and Social Networks*

Circle of friends, family changes, family network, marital / partnership status, attitude toward parental role, breastfeeding, childcare, language use, leisure and activities, parenting goals, parenting style, pregnancy, relationship to other parent or child

5. *Health and Care*

Alcohol consumption, health insurance, illness (self-reports of doctor diagnoses for sleep disorder, thyroid disorder, diabetes, asthma, cardiac disease, cancer, apoplectic stroke, migraine, high blood pressure, depression, dementia, joint disorder, chronic back problems, burnout, hypercholesterolemia, or other illness), reduced ability to work, sickness notifications to employer, smoking, state of health, stress and exhaustion, visits to the doctor, satisfaction with availability of care, health of child, physical and mental health of mother, nutrition, physical activity

6. *Home, Amenities, and Contributions of Private Households*

Childcare hours, leisure activities and costs, school attendance by child, change in residential situation, consumption, costs of housing, home ownership / rental, loans and mortgages, birth of children, number of books in the household, persons in household in need of care, pets, residential area, size and condition of home

7. *Education and Qualification*

Completed education and training, vocational training, educational aspirations for children, school enrollment of children

8. *Attitudes, Values, and Personality*

Affective well-being, Big Five personality traits, depressive traits, goals in life, impulsivity and patience, income justice, life satisfaction, lottery question, optimism/pessimism,

political tendency and orientation, reciprocity, religious affiliation, risk preference in different domains, satisfaction with various aspects, social responsibility, trust and fairness, wage justice, well-being aspects, worries, temperament of child

9. *Time Use and Environmental Behavior*

Time use for different activities, trip to work, use of transportation for different purposes

10. *Integration, Migration, Transnationalization*

Applying for German citizenship, disadvantage / discrimination based on ethnic origins, integration indicators, language skills, native language, regional attachment, sense of home

11. *Innovative Modules*

Anxiety and depression, assessment of contextualized emotions, risk attitudes, confusion, control strivings, dementia worry, determinants of ambiguity aversion, emotion regulation, expected financial market earnings, future life events, grit and entrepreneurship, happiness analyzer, impostor phenomenon, inattentional blindness, inequality attitudes, job preferences, job tasks, justice sensitivity, lottery play, multilingualism, narcissistic admiration and rivalry, ostracism, pension claims, perceived discrimination, physical attractivenes, self-control, self-evaluation and overconfidence in different life domains, sleep characteristics, smartphone usage, socio-economic effects of physical activity, status confidence and anxiety, subjective social status, work time preferences

In short, both pipelines filtered out 14 individuals whose reported sex did not match biological sex derived from genotype data as this could be an indication of genotyping error. The strict-QC pipeline excluded 260 individuals whose genotype missingness rate was more than 20% within any chromosome and 59 individuals with an excess number of heterozygote/homozygote genotypes, which could indicate genotyping errors. The mild-QC pipeline excluded only 36 individuals based on a per-chromosome missingness of more than 50% and 22 heterozygosity/homozygosity outliers. Using the mild-QC data, we identified 44 individuals of non-European ancestries, 25 of whom were available in the strict-QC sample. Before imputation, these individuals were excluded from the mild- and strict-QC samples.

We used the Haplotype Reference Consortium reference panel (r1.1) for imputation [37]. Imputation was completed for 2,497 individuals and 23,185,386 SNPs with imputation accuracy ($R^2$; a measure of how much confidence there is in the imputed genotype probabilities) greater than 0.1 in the mild-QC data, and 2,299 individuals and 22,201,548 SNPs with $R^2 > 0.1$ in the strict-QC data. Approximately 66% of the imputed SNPs are rare with minor allele frequencies (MAF) smaller than 0.01, and ∼24% of SNPs are common (MAF≥0.05; 5,463,110 in mild-QC, 5,463,110 in strict-QC). The average imputation accuracy in the mild-QC data is 0.664 and 0.695 in the strict-QC data. However, common SNPs (MAF≥0.05) are much more reliably imputed than rare SNPs, with an average imputation accuracy of 0.92 and 0.93 in the mild- and strict-QC data, respectively.

Using the imputed SNPs, we identified an additional 37 (2) individuals of non-European ancestries in the strict (mild) QC data on top of the 44 (25) individuals of non-European

ancestries excluded before imputation, respectively. Thus, $\sim 98\%$ of the genotyped SOEP-IS sample is of European ancestries (see section 4 in S1 File).

We constructed the first 20 principal components (PCs) of the genetic data for individuals with European ancestries based on $\sim 160,000$ approximately independent SNPs ($r^2 < 0.01$) with imputation accuracy $\geq 70\%$ and MAF $\geq 0.01$ (see section 5 in S1 File). We recommend using these genetic PCs in analyses as control variables for population stratification [38].

## Family relationship among genotyped participants

With the exemption of parent-offspring pairs, family relationships among the participants are only surveyed via their relationship with the head of the household. For the genotyped participants in the SOEP-IS across the available waves from 1998 to 2019, there are 877 reported relationships for the 602 household heads. The majority (515) of these relationships are with their spouse or partner, while 346 relationships are with their child (324 biological, 11 adopted or biological, and 11 stepchildren). The remaining relationships of household heads are with grandchildren (5), parents (4), a parent-in-law (1), a niece/nephew (3), a son/daughter-in-law (1), and a half-sibling (1).

By using the reported relationships to the household head as well as directly reported parent-child relationships, we inferred or found 609 parent-offspring, 142 full-sibling, and 17 second-degree relative pairs in the SOEP-G sample. In S1 Table, we compared these reported relationships to genetically inferred relationships obtained from KING [39]. We found that 19% of the pairs have inconsistencies between the reported and genetically inferred relationships. The deviations were mainly due to the low genotyping quality of some individuals. When considering only the individuals whose genotyping call rate was greater than 90% using directly genotyped SNPs, 92% of the pairs in the SOEP-G have consistent self-reported and genetic family relationships (see sections 3 and 6 of the S1 File for details). We found that most of the remaining inconsistencies are due to self-reported full siblings who are likely to be only half-siblings (13 out of 97 pairs). We also found 28 self-reported parent-child pairs that appear to be non-biological from 437 pairs in total.

Furthermore, focusing on the individuals with a genotype call rate greater than 90%, we identified 88 pairs whose family relationship information was unavailable in the survey data. These pairs consist of 7 parent-offspring, 19 full-siblings, 33 second-degree relatives, and 29 third or fourth-degree relative pairs.

Overall, out of 2,497 individuals, we genetically identified 703 individuals with at least one first-degree relative (parent-child or full sibling) and 728 individuals that have at least one relative with at least third-degree of relatedness (first cousins or great-grandparent-child). 1,769 individuals do not have close relatives based on the genetic data. Note that the related pairs reported here are not mutually exclusive, and some individuals can be related to multiple people.

## Polygenic indices

The effect sizes of individual single nucleotide polymorphisms (SNPs) on behavioral traits and complex diseases are usually tiny [40] ($R^2 < 0.05\%$). Polygenic indices (PGI) aggregate the effects of observed SNPs, weighting them by their estimated effect sizes from an independent genome-wide association study (GWAS) sample [36]. The predictive accuracy of a PGI depends on the GWAS sample size (+), the heritability of the trait (+), the number of causal genetic variants that influence the trait (-), and the extent to which the genetic architecture of the trait is similar across various environments and datasets (+) [41, 42]. Thanks to rapidly growing GWAS sample sizes in the past few years, the accuracy of PGIs has increased greatly, especially for individuals of European ancestries [16, 43]. PGIs are now beginning to capture a

**Table 3. Polygenic indices in the SOEP-G sample from single trait GWAS results.**

| Phenotype | # SNPs | GWAS N |
|---|---|---|
| Adventurousness [36, 44] | 1,147,160 | 557,923 |
| Age First Birth [36, 47] | 996,620 | 169,901 |
| Age First Menses (Women) [36, 48] | 1,142,133 | 309,043 |
| Alcohol Misuse [36, 49] | 1,145,324 | 120,684 |
| Alzheimer's* [50] | 1,115,709 | 455,258 |
| Any Ischemic Stroke* [50] | 850,822 | 446,696 |
| Any Stroke* [50] | 844,962 | 446,696 |
| Atrial Fibrillation* [50] | 850,822 | 1,030,836 |
| Asthma [36] | 1,159,334 | 418,164 |
| Asthma/Eczema/Rhinitis [36, 51] | 1,137,288 | 513,889 |
| Attention Deficit Hyperactivity Disorder (ADHD) [36, 52] | 1,083,048 | 57,386 |
| Body Mass Index (BMI) [36, 53] | 1,023,282 | 582,457 |
| Breast Cancer* [50] | 809,475 | 228,951 |
| Cannabis Use [36, 54, 55] | 1,087,000 | 156,756 |
| Cardioembolic Stroke* [50] | 844,996 | 446,696 |
| Childhood Reading [36] | 1,147,169 | 172,502 |
| Chronic Kidney Disease* [50] | 845,145 | 444,971 |
| Cigarettes per Day [36, 56] | 1,150,910 | 250,057 |
| Cognitive Performance [36, 57] | 1,148,362 | 222,914 |
| Depression* [50] | 835,515 | 500,199 |
| Depressive Symptoms [36, 58] | 1,138,362 | 619,272 |
| Diastolic Blood Pressure* [50] | 843,500 | 757,601 |
| Drinks per Week [36, 56] | 1,150,775 | 723,487 |
| Educational Attainment [21, 36] | 1,147,926 | 1,047,538 |
| Ever Smoker [36, 56] | 1,143,561 | 1,129,163 |
| Externalizing* [50] | 1,020,283 | 1,492,085 |
| Extraversion [36, 59, 60] | 1,113,746 | 73,906 |
| Hay Fever [36] | 1,159,334 | 403,179 |
| HDL Cholesterol* [50] | 847,159 | 187,167 |
| Height [36, 61] | 1,022,784 | 448,198 |
| Highest Math [21, 36] | 1,147,159 | 430,439 |
| Insomnia* [50] | 824,863 | 386,533 |
| Large Artery Stroke* [50] | 1,159,551 | 446,696 |
| Left Out of Social Activity [36] | 1,147,159 | 507,803 |
| Life Satisfaction: Family [36] | 1,159,202 | 141,864 |
| Life Satisfaction: Friends [36] | 1,159,184 | 138,807 |
| Longevity* [50] | 832,850 | 640,189 |
| Migraine [36, 62] | 1,146,834 | 421,013 |
| Morning Person [36, 63] | 1,123,260 | 362,840 |
| Narcissism [36] | 1,147,153 | 452,535 |
| Nearsightedness [36, 62] | 1,146,729 | 301,938 |
| Neuroticism [36, 59, 64] | 1,029,577 | 389,237 |
| Number Ever Born (Women) [36, 47] | 1,034,474 | 207,393 |
| Openness [36, 59, 65] | 987,746 | 72,308 |
| Physical Activity [36, 66] | 1,108,549 | 140,190 |
| Religious Attendance [36] | 1,159,336 | 383,466 |
| Risk Tolerance [36, 44] | 1,076,002 | 1,070,480 |

*(Continued)*

**Table 3.** (Continued)

| Phenotype | # SNPs | GWAS *N* |
|---|---|---|
| Schizophrenia* [50] | 829,801 | 105,318 |
| Self-Rated Health [36] | 1,144,515 | 911,102 |
| Self-Rated Math Ability [21, 36] | 1,147,159 | 564,692 |
| Small Vessel Stroke* [50] | 1,159,163 | 446,696 |
| Subjective Well-Being [36, 67] | 906,574 | 502,976 |
| Systolic Blood Pressure* [50] | 842,552 | 745,820 |
| Triglycerides* [50] | 847,159 | 177,861 |
| Type 2 Diabetes* [50] | 851,227 | 231,426 |

Notes: "# SNPs" is the number of SNPs that were used to construct the PGI.

"*" indicates PGIs for medical outcomes that were not originally included in Becker et al. 2021.

All 55 PGIs are constructed only for individuals of European ancestry (*N* = 2,495).

substantial part of the heritability of many traits, making them valuable for research in many scientific disciplines. For example, PGIs from the latest generation of GWAS analyses capture $\sim$12% of the variation in years of schooling [21], $\sim$10% of general cognitive ability [21], and up to 2% of various personality characteristics such as risk tolerance [44].

These PGIs are useful for follow-up analyses in samples much smaller than the original GWAS [16]. For example, a sample of *N* = 1,000 yields >90% statistical power to detect an association between a PGI and an outcome of interest if the PGI captures at least 1% of the phenotypic variation (two-sided *t*-test with α = 0.05). An association between an outcome and a PGI with $R^2$ = 10% can even be detected in a sample of only *N* = 110 individuals with 90% power.

We followed the methods used by Becker et al. 2021 [36] to create a repository of single- and multi-trait polygenic indices for 66 social-scientific and health traits for individuals of European ancestries in the SOEP-G sample. We used the largest currently available GWAS samples to create these PGIs, including publicly available GWAS summary statistics and non-publicly available GWAS results from 23andMe. We extended the list of 36 single-trait and 35 multi-trait PGIs in Becker et al. 2021 by including single-trait PGIs for 19 medical outcomes with well-powered GWAS summary statistics. The single-trait PGIs were based on univariate GWAS summary statistics (Table 3), whereas the multi-trait PGIs were based on multivariate MTAG analyses that exploit genetic correlations between several traits to improve predictive accuracy (S3 Table) [45].

Some of the PGIs that we created have corresponding phenotypes in the SOEP-G sample (e.g., educational attainment, height, BMI, risk tolerance), while others capture genetic predispositions for phenotypes that are not observable or incompletely measured (e.g., longevity, HDL cholesterol, blood pressure, and a variety of diseases including Alzheimer's, schizophrenia, stroke, atrial fibrillation, and breast cancer). These PGIs are useful proxies for unobserved traits and outcomes. For example, they can be used as control variables in studies that focus on environmental processes, such as socio-economic factors that influence health [17], to detect gene-environment interactions (e.g., heterogeneous responses to policy interventions) [5, 16], or as exogenously given proxies that do not change over the life course (e.g., to study genetic predisposition for health on labor market outcomes). Finally, the availability of genetic data and PGIs from parents and their children offers exciting, new ways to disentangle genetic and environmental channels of intergenerational transmission of health, behavior, and socioeconomic outcomes [3, 46].

## Results—What has been found?

The SOEP sample is currently used by more than 9,000 registered users from 54 countries [34]. About 300–400 publications annually are based on SOEP data, including OECD reports on international inequality development. Roughly 25% of these publications are in journals listed in the (social) science citation index, and more than 100 publications are based on SOEP-IS data. The SOEP is also an integral database for official government reports in Germany. Major research areas that include SOEP-based publications include life course development, inequality, mobility, psychological outcomes and attitudes, migration, transition to a unified Germany, and health. Thus, the SOEP data is widely used and provides an indispensable empirical foundation to describe longitudinal developments and relationships, and a better understanding of socioeconomic processes and behavior. It is a valuable resource to study associations between behavior, socioeconomic status, and health [23].

The genetic data in the SOEP-IS sample (SOEP-G) is a new addition to this valuable resource. We describe the first findings using the genetic data below.

### Predictive accuracy of polygenic indices for height, BMI, and educational attainment

Fig 3 shows the predictive accuracy of the PGIs for height and BMI in unrelated individuals from the SOEP-G sample, both for the mild and the strict version of the QC of the genetic data we carried out. We measure the predictive accuracy of the PGIs as the difference in the explained variance ($R^2$) before and after adding the PGI to a baseline regression that controls for a second-degree polynomial in the year of birth, sex, and their interactions, genotype batch indicators, and the top 20 genetic PCs. Since height and BMI were surveyed multiple times across waves, we first residualized height and BMI for age, age$^2$, sex, and their interactions within each wave and took the mean for each individual; then, as covariates, we used only genotype batch indicators and the top 20 genetic PCs. We obtained 95% confidence intervals by bootstrapping the sample 2,000 times.

Using this approach, the PGIs explain 22∼24% of the variance in height, 12∼13% of the variance in BMI, and 9% of the variance in educational attainment. Furthermore, the predictive accuracy was very similar for different levels of QC, which implies that the low genotyping quality in a part of the sample does not substantially reduce the predictive accuracy of the PGIs. Thus, researchers may to use the mild-QC version of the data for analyses using PGIs to take advantage of its ∼10% larger sample size and the corresponding gains in statistical power.

### Genetic and environmental correlations with height and BMI

We demonstrate the advantages of combining a representative population sample with genetic data by analyzing birth year cohort trends in body height and BMI over time. Specifically, we split the SOEP-G sample into PGI values below and above the median for height and BMI and plotted the average residualized phenotypic values after adjusting for sex in both groups for adults > = 20 years of age, binned into ten-year birth cohorts (Figs 4 and 5). Phenotypic values are residualized by regressing each observed phenotypic value on sex dummies using OLS. Each observation is assigned a residualized value representing the remaining variation in the phenotype that cannot be predicted by sex. Residualized values are then averaged by individuals across survey waves. The solid lines corresponding to the left axis report the average residualized values for each bin.

In the non-residualized data, individuals with high PGI values for height are, on average, 5.2 cm taller than those with low PGI height values (95% CI: 3.4–7.1cm). Fig 4 shows this

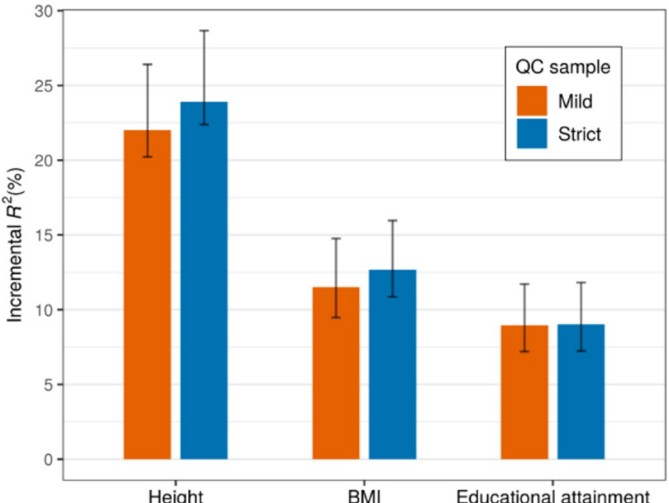

**Fig 3. Polygenic prediction in the SOEP-IS sample. Note**: The bars report the prediction accuracy of polygenic indices among unrelated individuals of European ancestries measured as incremental $R^2$. The sample size of the strict (mild) QC sample is 1,904 (2,094), 1,897 (2,086), and 1,857 (2,036) for height, BMI, and educational attainment, respectively. The error bars indicate 95% bootstrapped confidence intervals with 2,000 replications.

difference in average height by genetic predisposition is robust across birth year cohorts, reflecting a stable influence of the height PGI. Consistent with the previous findings [68], Fig 4 also demonstrates that younger birth cohorts are, on average, substantially taller than older birth cohorts. For example, individuals born in the 1923–1939 birth year cohort on average ($\sim$ 84 years old in the 2019 survey wave) are on average 6.6 cm shorter than those born in 1980–1999 birth year cohort (on average, $\sim$ 31 years old in the 2019 survey wave). This gain in

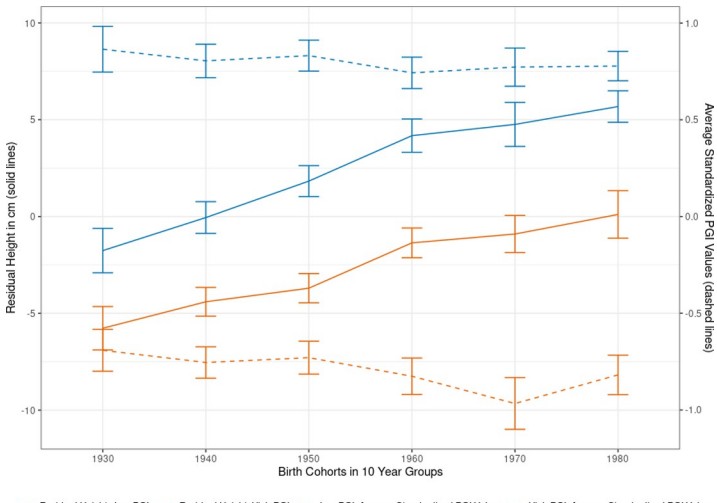

**Fig 4. Body height by birth cohorts and PGI values. Note**: Using the single-trait polygenic index (PGI) for body height, we split the sample of adults (older than 20 years) into two parts at the median PGI value (High PGI $N$ = 1,085; Low PGI: $N$ = 1,079). Self-reported height is residualized on sex and survey year before being averaged across survey waves. Each individual is assigned to a decadal cohort. Individuals born before between 1923 and 1939 are all in the 1930s cohort, while individuals born after 1980 are all in the 1980 group. Individuals born between 1940–1949, 1950–1959, 1960–1969, and 1970–1979 are labeled as 1940s, 1950s, 1960s, and 1970s respectively. We plotted the average observed residual height for each decadal cohort by PGI bin, along with 95% confidence intervals.

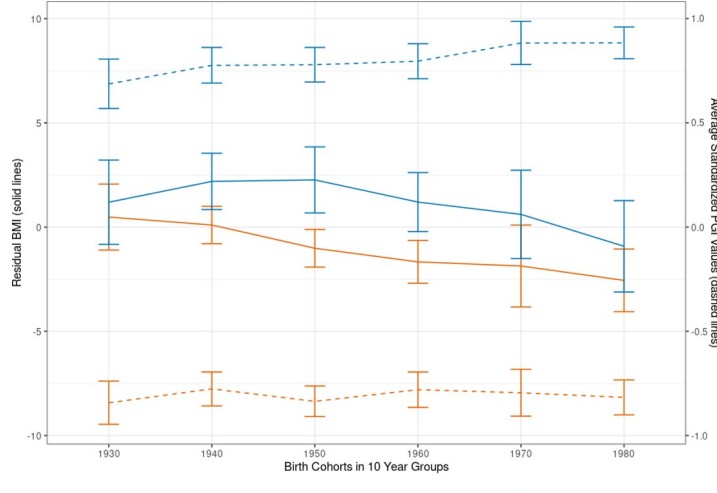

**Fig 5. Body mass index (BMI) by birth cohort and PGI values. Note**: Using the single-trait polygenic index (PGI) for BMI, we split the sample of adults (older than 20 years) into two parts at the median PGI value (High PGI: $N = 683$; Low PGI: $N = 775$). Self-reported BMI is residualized for sex and survey year before being averaged across survey waves. Each individual is assigned to a decadal cohort. Individuals born before between 1923 and 1939 are all in the 1930s cohort, while individuals born after 1980 are all in the 1980 group. Individuals born between 1940–1949, 1950–1959, 1960–1969, and 1970–1979 are labeled as 1940s, 1950s, 1960s, and 1970s respectively. We plotted the average observed residual BMI for each decadal cohort by PGI bin and 95% confidence intervals.

the average height of younger birth cohorts cannot be explained by observed genetic changes in the population. As Fig 4 shows through the dashed lines corresponding with the right *y*-axis, the average values of the (high and low) height PGI did not increase over time. Instead, the younger birth cohorts exhibit a slightly smaller PGI value than the older birth cohorts, possibly due to sample selection and mortality effects among older participants [69]. To disentangle potential age effects from birth cohort effects, S5 Table presents estimates from height regressed on the standardized height PGI, birth cohort dummies, including five-year age bin dummies. The results confirm a birth cohort effect on height that is separate from the genetic influences on height and aging effects. This implies that the substantial gains in the average height of the German population over time are partially due to improved environmental conditions, such as better nutrition and health care [70–72].

A similar analysis for BMI (Fig 5) shows that individuals with an above-median PGI have, on average, also a higher BMI (1.6 points higher for the High-PGI group in the non-residualized results, 95% CI 1.04–2.17). The heritability and the predictive accuracy of the PGI are lower for BMI than for height [2, 36]. Correspondingly, the average differences in BMI between the low and the high PGI group are not statistically significant for all birth year cohorts. Yet, similar to the analyses on height, we also observe birth cohort effects on BMI that cannot be explained by observed genetic variation in the BMI PGI. Individuals born in the youngest birth cohort (1980–1999, $\sim 31$ years old) have an average BMI of 2.3 points lower than those in the oldest birth cohort (1923–1939, $\sim 84$ years old). The higher BMI in the older birth cohorts is not due to observed genetic changes in the population over time. In fact, the average PGI is slightly lower in the older birth cohorts than in the younger ones, possibly due to sample selection and mortality effects among older participants [69]. S6 Table presents regression results from a robustness check that also included 5-year age bins as control variables, again confirming birth cohort effects that cannot be explained alone by aging or observed genetic variation. Thus, the higher BMI in the older birth cohorts is likely to be

caused by a combination of environmental effects such as differences in living conditions, socio-economic effects [73], or nutrition [74] in addition to general effects from aging.

## Polygenic indices as proxies for health

The broad set of PGIs we created is a valuable resource for research on inequalities in socioeconomic and health outcomes. Previous research has demonstrated that the genetic architectures of socioeconomic, behavioral, and health outcomes are often substantially overlapping [16, 40, 75]. This implies that PGIs for socio-economic or behavioral traits can also be proxies for health outcomes.

This is demonstrated in Fig 6 which presents the effect size from regressions of self-rated health on 28 single-trait PGIs (out of 55 tested single-trait PGIs overall) whose estimated standardized coefficients are greater than |±0.1| All regressions controlled for five-year age bins, sex, and their interactions, and the first 20 genetic principal components. 18 PGIs are statistically distinguishable from zero after a Bonferonni correction for 55 tested hypotheses (marked with *).

We find positive associations between self-rated health and PGIs for self-rated health, age at first birth, educational attainment, subjective well-being, highest math class taken, religious attendance, longevity, cognitive performance, physical activity, self-rated math ability, and age at first menses. Furthermore, we find negative health correlations of the PGIs for externalizing, depression, ADHD, number of children ever born, insomnia, neuroticism, smoking, and being left out of social activities—all of which are PGIs for behavioral, social, or cognitive phenotypes. Moreover, the PGIs for BMI, high blood pressure, type 2 diabetes, large artery stroke, triglycerides, and asthma all have the expected negative correlations with self-rated health.

## Discussion

### What are the main strengths and weaknesses?

Major strengths of the SOEP-G data include:

i. the sample selection, which yields the only currently genotyped sample that is representative of the entire population in Germany;

ii. the longitudinal nature of the data with annual observations since 2011 (for a subset of individuals and phenotypes, annual observations even go back to 1998);

iii. the rich questionnaire content, including self-reported health outcomes and detailed information on socio-economic status, living conditions, family dynamics, personality, preferences and attitudes is another major strength of the data;

iv. the possibility to use detailed geo-coding, standardized occupation codes, and links to external databases such as the German Pension Insurance and the Employer-Employee Study;

v. the broad set of state-of-the-art polygenic indices that we created, which lower the entry barriers for researchers to use genetically informed study designs;

vi. the continuing annual collection of data that also allows researchers to integrate new survey modules, biomarkers, and experiments in the future by following the application procedures of the SOEP-IS management team [22];

vii. the household sampling procedure that collects data on all family members. The SOEP-G sample contains 501 parent-offspring pairs, 152 parent-offspring trios, 107 full-siblings,

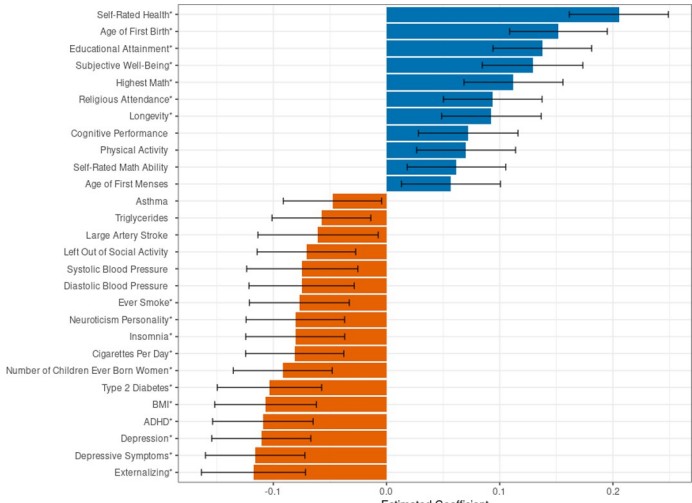

**Fig 6. Associations between polygenic indices and self-rated health. Note**: Analyses in the SOEP-G sample, $N = 2,060$. Self-rated health is measured by a 5-point Likert scale where a 1 indicates poor health and a 5 indicates very good health. Each self-rated health observation is regressed on five-year age-bin dummies, sex dummies, and the interaction of sex and age-bin dummies. We take the estimated residual from the previous regression, compute the average residual value for each individual, and regress each PGI along with 20 genetic principal components on these residuals where each individual has one observation. The estimated standardized betas from each PGI are reported in the figure. The figure represents 28 single-trait PGIs with an effect size greater than $|±0.1|$, out of 55 single-trait PGIs overall. PGIs marked with an * are statistically distinguishable from zero after a Bonferonni correction. Error bars represent a 95% confidence interval around the estimated beta for each PGI.

and 12 second-degree relatives (including half-siblings) with matching self-reported and genetically inferred relationships. This data structure enables genetically informed studies on a wide range of research topics, including the intergenerational transmission of inequalities in health and well-being, as well as studies that identify how environmental factors such as parenting style influence the developmental trajectory of children and youths;

viii. the availability of epigenetic data that in the near future will be added for a substantial part of the SOEP-G sample, thus opening further research opportunities on the associations between social environment and physical health;

ix. the possibility to extend the collection of genetic data to all SOEP surveys, thus substantially increasing the available sample size for genetically informed analyses.

Compared to other datasets that were included in the Polygenic Index (PGI) Repository of the SSGAC [36], the SOEP-G is the only German sample, and it has the broadest coverage of social scientific outcomes, many of which have been repeatedly collected over time. Although the sample size of the SOEP-G is larger than several other studies included in the PGI Repository (e.g. Dunedin, E-Risk, Texas Twins), we still caution that researchers using the data should pay attention to statistical power in their analyses. In particular, the sample size may be too limited for analyses of single genetic variants or sub-parts of the sample (e.g., specific age groups or geographic areas). A further limitation is that a part of the sample (19%) did not pass the strict quality control thresholds of genetic data that are usually employed in genetic epidemiology (call rates > 95%). However, our mild-QC pipeline enables well-performing PGIs in 2,495 individuals (96% of the successfully genotyped sample).

Another possible limitation is that the currently available health outcomes are limited in detail and based on self-reports rather than detailed digital health records.

Future expansions of the collected health data, e.g., digital health records or extended health surveys conducted by trained medical professionals, would further increase the utility of the SOEP samples for epidemiological research. Furthermore, the resolution and completeness of the collected genetic data could be improved further, e.g., by high-throughput sequencing methods. We have stored residual DNA samples for this purpose. Access to those samples can be requested via DIW Berlin.

## Opportunities for future research

The genetic data we added to the SOEP-IS sample opens a broad range of opportunities for future research. For example, social scientists and economists who are studying the effects of environmental or policy changes on behavior or socio-economic outcomes can now use the PGIs we constructed in the SOEP-IS to control for otherwise unobservable genetic confounds (e.g., to estimate the returns to schooling) [12, 17] and to detect gene-environment interactions (e.g., heterogeneous responses to policy interventions such as changes in tobacco taxes on smoking behavior) [5, 16]. The PGIs can also be used as exogenously given proxies for outcomes that do not change over the life course (e.g., to study genetic predisposition for health on labor market outcomes) or as proxies for outcomes that are not observed in the SOEP-IS data otherwise (e.g., blood pressure and triglycerides levels).

Furthermore, the family data structure in the SOEP-IS, in combination with PGIs, enables new ways to study intergenerational transmission of inequalities in health and well-being as well as studies that identify how environmental factors such as parenting style influence the developmental trajectory of children [3, 76].

## Supporting information

**S1 Fig. Histogram of minor allele frequencies of genotyped autosomal biallelic SNPs.**
(PDF)

**S2 Fig. Histogram of genotype call rates of SNPs and samples.**
(PDF)

**S3 Fig. Binned scatter plot of SNP missing rates over minor allele frequencies.**
(PDF)

**S4 Fig. Histogram of mean sample call rates by interviewer.**
(PDF)

**S5 Fig. Homozygosity / heterozygosity outliers.**
(PDF)

**S6 Fig. Pre-imputation (mild-)QC—Ancestry filtering.**
(PDF)

**S7 Fig. Imputed data MAF distribution.**
(PDF)

**S8 Fig. Imputation accuracy distribution.**
(PDF)

**S9 Fig. Mean imputation accuracy by MAF.**
(PDF)

**S10 Fig. Post-imputation mild-QC—Ancestry filtering.**
(PDF)

**S11 Fig. Post-imputation strict-QC—Ancestry filtering.**
(PDF)

**S12 Fig. Flow-chart quality control of genetic data.**
(PDF)

**S1 Table. Comparison of reported and genetically inferred family relationships.**
(XLSX)

**S2 Table. Polygenic prediction accuracy for height, BMI, and educational attainment.**
(XLSX)

**S3 Table. Multi-trait polygenic indices in the Gene-SOEP sample.**
(XLSX)

**S4 Table. Predictive accuracy of polygenic indices on self-rated health.**
(XLSX)

**S5 Table. Observed height birth cohort effect estimates.**
(XLSX)

**S6 Table. Observed BMI birth cohort effect estimates.**
(XLSX)

**S7 Table. Correlation between child EA PGI and parental characteristics conditioning on parental PGIs.**
(XLSX)

**S1 File. Supporting information for Cohort profile: Genetic data in the German Socio-Economic Panel Innovation Sample (Gene-SOEP).**
(PDF)

## Acknowledgments

We are deeply indebted to all individuals who have agreed to participate in the German Socio-Economic Panel Innovation Survey. The construction of polygenic indices in this study was made possible by the generous public sharing of summary statistics from published GWAS from many research consortia. We would like to thank the studies that made these consortia possible, the researchers involved, and the participants in those studies, without whom this effort would not be possible. We would also like to thank the research participants and employees of 23andMe for making this work possible.

## Author Contributions

**Conceptualization:** Philipp D. Koellinger, Gert Wagner, Ralph Hertwig.

**Data curation:** Philipp D. Koellinger, Hyeokmoon Kweon, Jan Goebel, David Richter, Lisa Reiber, Bettina Maria Zweck, Gert Wagner, Ralph Hertwig.

**Formal analysis:** Philipp D. Koellinger, Aysu Okbay, Hyeokmoon Kweon, Annemarie Schweinert, David Richter, Lisa Reiber, Bettina Maria Zweck.

**Funding acquisition:** Philipp D. Koellinger, Daniel W. Belsky, Pietro Biroli, Rui Mata, Elliot M. Tucker-Drob, K. Paige Harden, Gert Wagner, Ralph Hertwig.

**Project administration:** Philipp D. Koellinger, Gert Wagner, Ralph Hertwig.

**Software:** Richard Karlsson Linnér.

**Supervision:** Philipp D. Koellinger, Gert Wagner, Ralph Hertwig.

**Validation:** Philipp D. Koellinger.

**Visualization:** Hyeokmoon Kweon, Annemarie Schweinert, Jan Goebel, David Richter.

**Writing – original draft:** Philipp D. Koellinger, Aysu Okbay, Hyeokmoon Kweon, Annemarie Schweinert, Richard Karlsson Linnér, Jan Goebel, David Richter, Lisa Reiber, Bettina Maria Zweck, Daniel W. Belsky, Pietro Biroli, Rui Mata, Elliot M. Tucker-Drob, K. Paige Harden, Gert Wagner, Ralph Hertwig.

**Writing – review & editing:** Philipp D. Koellinger, Aysu Okbay, Hyeokmoon Kweon, Annemarie Schweinert, Richard Karlsson Linnér, Jan Goebel, David Richter, Lisa Reiber, Bettina Maria Zweck, Daniel W. Belsky, Pietro Biroli, Rui Mata, Elliot M. Tucker-Drob, K. Paige Harden, Gert Wagner, Ralph Hertwig.

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
