## [Decision Letter · Decision Letter 0]

4 Jul 2023

PONE-D-23-05350Cohort Profile: Genetic data in the German Socio-Economic Panel Innovation Sample (SOEP-G)PLOS ONE

Dear Dr. Koellinger,

Thank you for submitting your manuscript to PLOS ONE. After careful consideration, we feel that it has merit but does not fully meet PLOS ONE’s publication criteria as it currently stands. Therefore, we invite you to submit a revised version of the manuscript that addresses the points raised during the review process.

We look forward to receiving your revised manuscript.

Kind regards,

José Alberto Molina

Academic Editor

PLOS ONE

Journal Requirements:

3. Please expand the acronym “NIH/NICHD” (as indicated in your financial disclosure) so that it states the name of your funders in full.

6. We note that Figure 2 in your submission contain map images which may be copyrighted. All PLOS content is published under the Creative Commons Attribution License (CC BY 4.0), which means that the manuscript, images, and Supporting Information files will be freely available online, and any third party is permitted to access, download, copy, distribute, and use these materials in any way, even commercially, with proper attribution. For these reasons, we cannot publish previously copyrighted maps or satellite images created using proprietary data, such as Google software (Google Maps, Street View, and Earth). For more information, see our copyright guidelines: http://journals.plos.org/plosone/s/licenses-and-copyright.

(1) You may seek permission from the original copyright holder of Figure 2 to publish the content specifically under the CC BY 4.0 license.  

Reviewers' comments:

Reviewer's Responses to Questions

**Comments to the Author**

1. Is the manuscript technically sound, and do the data support the conclusions?

Reviewer #1: Yes

Reviewer #2: Yes

2. Has the statistical analysis been performed appropriately and rigorously? 

Reviewer #1: Yes

Reviewer #2: Yes

3. Have the authors made all data underlying the findings in their manuscript fully available?

Reviewer #1: Yes

Reviewer #2: Yes

4. Is the manuscript presented in an intelligible fashion and written in standard English?

Reviewer #1: Yes

Reviewer #2: Yes

5. Review Comments to the Author

Reviewer #1: This paper highlights the wealthof the SOEP data to conduct health and genetic studies. Specifically, the authors emphasize the genetic data collectedin the SOEP. I think that the article is very well written and suitable for publication, but some minor corrections should be made before publication. Specifically, when I come across the manuscript, I do not see the motivation behind this work. I think it is important to improve it, by pointing out the specific information gathered in the SOEP-IS sample, in order to serve as a "guide" for future research. Furthermore, the contribution of the paper is not described. Finally, possible future research lines, taking advantage of this dataset knowledge, should be pointed out, for both practitioners and researchers from different disciplines: economics, medical, health, social sciences,...

In Section 2 (list all Sections/Sub-sections), describe in more detail the GSOEP data, in general terms. The focus of the survey, the target respondent (individual vs. household), the panel data structure, the time period availaible at the time of writing the article,...

In Page 9, I would prefer to include a Table and show both mean and SD of the SOEP-G and Census sample statistics. It would be more visible to readers to identify potential differences between SOEP-G and census data.

In the last Section, you could suggest possible improvements of the health and genetic measures collected in this survey for future waves.

Reviewer #2: This paper, which details the genetic data collection for approximately 2,600 individuals in the German Socio-Economic Panel, is well written and provides an invaluable resource for researchers. Those investigating sociodemographic and economic questions using genetic data will find this particularly beneficial.

However, I have two suggestions to improve the paper further:

The authors note in their discussion of Figure 4 that "younger birth cohorts are on average substantially taller than older birth cohorts". It would add weight to their observation if they referenced previous studies that have also found this, such as the one available at https://www.ncbi.nlm.nih.gov/pmc/articles/PMC2809930/, which highlights the role of the socioeconomic environment (namely, income and disease).

This paper is clear, well-written, and I believe will be highly cited, offering considerable assistance to the research community across various fields in utilizing genetic data in demographic and socioeconomic research. Perhaps this can be emphasized in the paper, with examples of recent research such as those available at the following links:

https://pubmed.ncbi.nlm.nih.gov/32587483/

https://www.journals.uchicago.edu/doi/abs/10.1086/705415

https://www.sciencedirect.com/science/article/pii/S0927537121000580

Additionally, I identified a few minor typographical errors, and I recommend the authors proofread the article to find any others:

p.14: .28.

p.15: The genotype missingness rate was greater than 5% in 484 individuals.

p.16: indication of genotyping error

6. PLOS authors have the option to publish the peer review history of their article (what does this mean?). If published, this will include your full peer review and any attached files.

Reviewer #1: No

Reviewer #2: No

---

## [Author Response · Author response to Decision Letter 0]

17 Oct 2023

Journal requirements

1. Please ensure that your manuscript meets PLOS ONE's style requirements, including

those for file naming. The PLOS ONE style templates can be found at.

We implemented style instructions. Please let us know if any additional changes need to be made.

2. We note that the grant information you provided in the ‘Funding Information’ and

‘Financial Disclosure’ sections do not match. When you resubmit, please ensure that you provide the correct grant numbers for the awards you received for your study in the ‘Funding Information’ section.

We listed the correct grant numbers (as far as available) and funding information in the “Funding Information” section of the online submission system. 

3. Please expand the acronym “NIH/NICHD” (as indicated in your financial disclosure) so

that it states the name of your funders in full. This information should be included in your cover letter; we will change the online submission form on your behalf.

We expanded the NIH/NICHD acronym and included the updated financial disclosure statement in the cover letter.

4. In your Data Availability statement, you have not specified where the minimal data set

underlying the results described in your manuscript can be found. PLOS defines a study's

minimal data set as the underlying data used to reach the conclusions drawn in the

manuscript and any additional data required to replicate the reported study findings in their

entirety. All PLOS journals require that the minimal data set be made fully available. For

more information about our data policy, please see http://journals.plos.org/plosone/s/dataavailability.

Upon re-submitting your revised manuscript, please upload your study’s minimal

underlying data set as either Supporting Information files or to a stable, public repository

and include the relevant URLs, DOIs, or accession numbers within your revised cover

letter. For a list of acceptable repositories, please see

http://journals.plos.org/plosone/s/data-availability#loc-recommended-repositories. Any

potentially identifying patient information must be fully anonymized.

Important: If there are ethical or legal restrictions to sharing your data publicly, please

explain these restrictions in detail. Please see our guidelines for more information on what

we consider unacceptable restrictions to publicly sharing data:

http://journals.plos.org/plosone/s/data-availability#loc-unacceptable-data-accessrestrictions.

Note that it is not acceptable for the authors to be the sole named individuals

responsible for ensuring data access.

We will update your Data Availability statement to reflect the information you provide in

your cover letter.

We are in the process of making the raw genetic data available via the European Genome-Phenome Archive. We will make the URL, accession number, or DOI available to you before publication. 

As mentioned in the Data Availability statement, DIW Berlin shares the genetic principal component and all polygenic indices constructed in a standard

phenotype file (contact email soepmail@diw.de). We have uploaded this minimal dataset as a Supporting Information file for the reviewers and editors. Please note that this dataset should NOT be published in this manuscript because the file contains sensitive personal information. Researchers who wish to access that data must apply (https://www.diw.de/en/diw_01.c.601584.en/data_access.html). 

5. We note that you have stated that you will provide repository information for your data

at acceptance. Should your manuscript be accepted for publication, we will hold it until

you provide the relevant accession numbers or DOIs necessary to access your data. If you

wish to make changes to your Data Availability statement, please describe these changes in

your cover letter and we will update your Data Availability statement to reflect the

information you provide.

We updated the Data Availability statement in the online submission statement and added it to the cover letter.

6. We note that Figure 2 in your submission contain map images which may be

copyrighted. All PLOS content is published under the Creative Commons Attribution

License (CC BY 4.0), which means that the manuscript, images, and Supporting

Information files will be freely available online, and any third party is permitted to access,

download, copy, distribute, and use these materials in any way, even commercially, with

proper attribution. For these reasons, we cannot publish previously copyrighted maps or

satellite images created using proprietary data, such as Google software (Google Maps,

Street View, and Earth). For more information, see our copyright guidelines:

http://journals.plos.org/plosone/s/licenses-and-copyright.

Figure 2 satisfies your requirements. We added the following note to Figure 2: 

“Own illustration based on spatial datasets from Reference [29] under data license Germany – attribution – Version 2.0 dl-de/by20 (https://www.govdata.de/dl-de/by-2-0)” 

Reference [29]: 

Federal Agency for Catography and Geodesy. Administrative areas Germany 1:250,000 as of 31.12. In: Bundesamt fuer Katographie und Geodaesie [Internet]. 31 Dec 2021. Available: https://gdz.bkg.bund.de/index.php/default/digitale-geodaten/verwaltungsgebiete/verwaltungsgebiete-1-250-000-stand-31-12-vg250-31-12.html

And here is the text from https://www.govdata.de/dl-de/by-2-0 describing the dl-de/by20 license:

“Data licence Germany – attribution – version 2.0

(1) Any use will be permitted provided it fulfils the requirements of this "Data licence Germany – attribution – Version 2.0".

The data and meta-data provided may, for commercial and non-commercial use, in particular

be copied, printed, presented, altered, processed and transmitted to third parties;

be merged with own data and with the data of others and be combined to form new and independent datasets;

be integrated in internal and external business processes, products and applications in public and non-public electronic networks.

(2) The user must ensure that the source note contains the following information:

the name of the provider,

the annotation "Data licence Germany – attribution – Version 2.0" or "dl-de/by-2-0" referring to the licence text available at www.govdata.de/dl-de/by-2-0, and

a reference to the dataset (URI).

This applies only if the entity keeping the data provides the pieces of information 1-3 for the source note.

(3) Changes, editing, new designs or other amendments must be marked as such in the source note.”

7. Please review your reference list to ensure that it is complete and correct. If you have

cited papers that have been retracted, please include the rationale for doing so in the

manuscript text, or remove these references and replace them with relevant current

references. Any changes to the reference list should be mentioned in the rebuttal letter that

accompanies your revised manuscript. If you need to cite a retracted article, indicate the

article’s retracted status in the References list and also include a citation and full reference

for the retraction notice.

We rechecked our references. To the best of our knowledge, they are complete and correct and do not contain retracted publications. 

Reviewer #1

This paper highlights the wealth of the SOEP data to conduct health and

genetic studies. Specifically, the authors emphasize the genetic data collected in the SOEP.

I think that the article is very well written and suitable for publication, but some minor

corrections should be made before publication. Specifically, when I come across the

manuscript, I do not see the motivation behind this work. I think it is important to improve

it, by pointing out the specific information gathered in the SOEP-IS sample, in order to

serve as a "guide" for future research. Furthermore, the contribution of the paper is not

described. Finally, possible future research lines, taking advantage of this dataset

knowledge, should be pointed out, for both practitioners and researchers from different

disciplines: economics, medical, health, social sciences,...

Thank you for your comments! 

Our primary motivation for collecting genetic data in the SOEP-IS sample was to create additional research opportunities and further enhance the value of this rich, longitudinal, population-based dataset. The contribution of the paper is to describe the newly collected genetic data and to offer first results afforded by them. 

We clarified this in the rewritten abstract, which now reads as follows:

“The German Socio-Economic Panel (SOEP) serves a global research community by providing representative annual longitudinal data of respondents living in private households in Germany. The dataset offers a valuable life course panorama, encompassing living conditions, socioeconomic status, familial connections, personality traits, values, preferences, health, and well-being. To amplify research opportunities further, we have extended the SOEP Innovation Sample (SOEP-IS) by collecting genetic data from 2,598 participants, yielding the first genotyped dataset for Germany based on a representative population sample (SOEP-G). The sample includes 107 full-sibling pairs, 501 parent-offspring pairs, and 152 triads, which overlap with the parent-offspring pairs. Leveraging the results from well-powered genome-wide association studies, we created a repository comprising 66 polygenic indices (PGIs) in the SOEP-G sample. We show that the PGIs for height, BMI, and educational attainment capture 22∼24%, 12∼13%, and 9% of the variance in the respective phenotypes. Using the PGIs for height and BMI, we demonstrate that the considerable increase in average height and the decrease in average BMI in more recent birth cohorts cannot be attributed to genetic shifts within the German population or to age effects alone. These findings suggest an important role of improved environmental conditions in driving these changes. Furthermore, we show that higher values in the PGIs for educational attainment and the highest math class are associated with better self-rated health, illustrating complex relationships between genetics, cognition, behavior, socio-economic status, and health. In summary, the SOEP-G data and the PGI repository we created provide a valuable resource for studying individual differences, inequalities, life-course development, health, and interactions between genetic predispositions and the environment.”

Pages 4-6 in the introduction describe our motivation for this work in greater detail. In particular:

“While genetically informed study designs are already common in medical research and have yielded numerous important insights into disease mechanisms, the use of genetic data in the behavioral and social sciences is still relatively rare.16 Nevertheless, integrating genetic data into the research of the behavioral and social sciences (e.g., economics, psychology, sociology, political science) opens up new possibilities to 

(i) control for genetic confounders that are otherwise unknown and that may lead to biased empirical results,

(ii) increase the statistical power of empirical analyses by absorbing residual variance in multiple regression analyses, yielding smaller standard errors of the estimated parameters, 

(iii) study the interactions of genetic factors and environmental exposures, 

(iv) use random genetic differences among individuals to identify causal pathways, and

(v) better understand how social (dis)advantages are transmitted across generations and how parents, peers, teachers, and policymakers can potentially alleviate or amplify such (dis)advantages.

Thus, integrating genetic data into the behavioral and social sciences offers researchers new tools to study key questions in their fields to reach more robust inference on the basis of their empirical analyses.

…

These considerations motivated us to collect genetic data in the Innovation Sample of the German Socio-Economic Panel Study (SOEP-IS), with the goal of contributing additional value to the already existing and widely known interdisciplinary and longitudinal SOEP data set that is accessible and frequently used by the global scientific community. The addition of genetic data to this sample opens up many new research opportunities for both the medical and the social-science research community.”

We have also added examples for possible future research lines in the article's discussion section: 

“Opportunities for future research

The genetic data we added to the SOEP-IS sample opens a broad range of opportunities for future research. For example, social scientists and economists who are studying the effects of environmental or policy changes on behavior or socio-economic outcomes can now use the PGIs we constructed in the SOEP-IS to control for otherwise unobservable genetic confounds (e.g., to estimate the returns to schooling) and to detect gene-environment interactions (e.g., heterogeneous responses to policy interventions such as changes in tobacco taxes on smoking behavior). The PGIs can also be used as exogenously given proxies for outcomes that do not change over the life course (e.g., to study genetic predisposition for health on labor market outcomes) or as proxies for outcomes that are not observed in the SOEP-IS data otherwise (e.g., blood pressure and triglycerides levels). 

Furthermore, the family data structure in the SOEP-IS, in combination with PGIs, enables new ways to study intergenerational transmission of inequalities in health and well-being as well as studies that identify how environmental factors such as parenting style influence the developmental trajectory of children.”

In Section 2 (list all Sections/Sub-sections), describe in more detail the GSOEP data, in general terms. The focus of the survey, the target respondent (individual vs. household), the panel data structure, the time period availaible at the time of writing the article,...

We describe the focus of the GSOEP data, the sampling procedure, the panel data structure and the available time periods on pages 6-17. Figure 1 illustrates the lifecourse perspective of the sample. Figure 2 shows the geographic distribution of genotyped participants. Box 1 provides an overview of the available variables. On p. 15, we refer to the online companion for the entire data collection (http://companion-is.soep.de/), which provides a full overview of all collected variables, survey design, questionnaires, and the target population. Tables 1 and 2 provide descriptive statistics of the sample. We also included references to articles that describe the SOEP and the SOEP-IS samples in even greater detail:

[22] Richter D, Schupp J. The SOEP Innovation Sample (SOEP IS). Journal of Contextual Economics – Schmollers Jahrbuch. 2015;135: 389–400.

[23] Goebel J, Grabka MM, Liebig S, Kroh M, Richter D, Schröder C, et al. The German Socio-Economic Panel (SOEP). Jahrbücher für Nationalökonomie und Statistik (Journal of Economics and Statistics). 2019.

In Page 9, I would prefer to include a Table and show both mean and SD of the SOEP-G and Census sample statistics. It would be more visible to readers to identify potential

differences between SOEP-G and census data.

Our manuscript contains a comparison of our sample with the census with gender, age, and region in the text on p. 9. We agree that comparisons for other variables would be desirable. Unfortunately, that is not possible. One reason is that there are no published figures from the census that we can easily fall back on. An analysis of the microdata would be required, which we do not have access to. The other reason is that definitions of variables in the SOEP and the census (e.g. for employment) do not match perfectly (the devil is in the details) or that the census does not collect those data (e.g. smoking, drinking). The team at DIW Berlin has already spent months trying to accomplish a broader comparison of the SOEP data and the census, unfortunately without success.

In the last Section, you could suggest possible improvements of the health and genetic

measures collected in this survey for future waves.

We added the following text on p. 32:

“Future expansions of the collected health data, e.g., digital health records or extended health surveys conducted by trained medical professionals, would further increase the utility of the SOEP samples for epidemiological research. Furthermore, the resolution and completeness of the collected genetic data could be improved further, e.g., by high-throughput sequencing methods. We have stored residual DNA samples for this purpose. Access to those samples can be requested via DIW Berlin.”

Reviewer #2

This paper, which details the genetic data collection for approximately 2,600 individuals in the German Socio-Economic Panel, is well written and provides an

invaluable resource for researchers. Those investigating sociodemographic and economic questions using genetic data will find this particularly beneficial.

Thank you very much for your positive and constructive comments!

However, I have two suggestions to improve the paper further:

The authors note in their discussion of Figure 4 that "younger birth cohorts are on average substantially taller than older birth cohorts". It would add weight to their observation if they referenced previous studies that have also found this, such as the one available at https://www.ncbi.nlm.nih.gov/pmc/articles/PMC2809930/, which highlights the role of the socioeconomic environment (namely, income and disease).

Thank you for this great reference! We now cite this paper in the discussion of Figure 4. We also highlighted this finding in the revised abstract.

This paper is clear, well-written, and I believe will be highly cited, offering considerable assistance to the research community across various fields in utilizing genetic data in

demographic and socioeconomic research. Perhaps this can be emphasized in the paper, with examples of recent research such as those available at the following links:

https://pubmed.ncbi.nlm.nih.gov/32587483/

https://www.journals.uchicago.edu/doi/abs/10.1086/705415

https://www.sciencedirect.com/science/article/pii/S0927537121000580

Thank you. We now cite these three studies on p. 4:

“Thus, integrating genetic data into the behavioral and social sciences offers researchers new tools to study key questions to reach more robust inferences based on their empirical analyses, as illustrated by several recent examples [18–20].”

Additionally, I identified a few minor typographical errors, and I recommend the authors

proofread the article to find any others:

p.14: .28.

Fixed (this was a reference that was wrongly formatted)

p.15: The genotype missingness rate was greater than 5% in 484 individuals.

This was not a typo, unfortunately. The text following this statement and a substantial part of the Supplementary Information talk about possible reasons for this and how we addressed this issue. 

p.16: indication of genotyping error

Fixed.

We have carefully proofread the article again and made minor grammatical changes throughout the text.

---

## [Decision Letter · Decision Letter 1]

13 Nov 2023

Cohort Profile: Genetic data in the German Socio-Economic Panel Innovation Sample (SOEP-G)

PONE-D-23-05350R1

Dear Dr. Koellinger,

We’re pleased to inform you that your manuscript has been judged scientifically suitable for publication and will be formally accepted for publication once it meets all outstanding technical requirements.

Kind regards,

José Alberto Molina

Academic Editor

PLOS ONE

Additional Editor Comments (optional):

Reviewers' comments:

Reviewer's Responses to Questions

**Comments to the Author**

1. If the authors have adequately addressed your comments raised in a previous round of review and you feel that this manuscript is now acceptable for publication, you may indicate that here to bypass the “Comments to the Author” section, enter your conflict of interest statement in the “Confidential to Editor” section, and submit your "Accept" recommendation.

Reviewer #1: All comments have been addressed

Reviewer #2: All comments have been addressed

2. Is the manuscript technically sound, and do the data support the conclusions?

Reviewer #1: Yes

Reviewer #2: Yes

3. Has the statistical analysis been performed appropriately and rigorously? 

Reviewer #1: Yes

Reviewer #2: Yes

4. Have the authors made all data underlying the findings in their manuscript fully available?

Reviewer #1: Yes

Reviewer #2: Yes

5. Is the manuscript presented in an intelligible fashion and written in standard English?

Reviewer #1: Yes

Reviewer #2: Yes

6. Review Comments to the Author

Reviewer #1: I have no additional comments. The authors have addressed all my suggestions and the paper is suitable for publication in the journal.

Reviewer #2: Thank you for your revisions. I have no further comments. In my opinion, the paper is ready to be published.

7. PLOS authors have the option to publish the peer review history of their article (what does this mean?). If published, this will include your full peer review and any attached files.

Reviewer #1: No

Reviewer #2: No

---

## [Editor Report · Acceptance letter]

17 Nov 2023

PONE-D-23-05350R1 

Cohort Profile:
Genetic data in the German Socio-Economic Panel Innovation Sample (SOEP-G) 

Dear Dr. Koellinger:

I'm pleased to inform you that your manuscript has been deemed suitable for publication in PLOS ONE. Congratulations! Your manuscript is now with our production department. 

Kind regards, 

on behalf of

Professor José Alberto Molina 

Academic Editor

PLOS ONE